

# Analyzing mixing systems using a new generation of Bayesian tracer mixing models

Brian C. Stock[1], Andrew L. Jackson[2], Eric J. Ward[3], Andrew C. Parnell[4], Donald L. Phillips[5] and Brice X. Semmens[1]

[1] Scripps Institution of Oceanography, University of California, San Diego, La Jolla, CA, USA
[2] Department of Zoology, School of Natural Sciences, University of Dublin, Trinity College, Dublin, Ireland
[3] Northwest Fisheries Science Center, National Marine Fisheries Service, National Oceanic and Atmospheric Administration, Seattle, WA, USA
[4] School of Mathematics and Statistics, Insight Centre for Data Analytics, University College Dublin, Dublin, Ireland
[5] EcoIsoMix.com, Corvallis, OR, USA

Corresponding authors
Brian C. Stock, b1stock@ucsd.edu
Brice X. Semmens,
bsemmens@ucsd.edu

## ABSTRACT

The ongoing evolution of tracer mixing models has resulted in a confusing array of software tools that differ in terms of data inputs, model assumptions, and associated analytic products. Here we introduce MixSIAR, an inclusive, rich, and flexible Bayesian tracer (e.g., stable isotope) mixing model framework implemented as an open-source R package. Using MixSIAR as a foundation, we provide guidance for the implementation of mixing model analyses. We begin by outlining the practical differences between mixture data error structure formulations and relate these error structures to common mixing model study designs in ecology. Because Bayesian mixing models afford the option to specify informative priors on source proportion contributions, we outline methods for establishing prior distributions and discuss the influence of prior specification on model outputs. We also discuss the options available for source data inputs (raw data versus summary statistics) and provide guidance for combining sources. We then describe a key advantage of MixSIAR over previous mixing model software—the ability to include fixed and random effects as covariates explaining variability in mixture proportions and calculate relative support for multiple models via information criteria. We present a case study of *Alligator mississippiensis* diet partitioning to demonstrate the power of this approach. Finally, we conclude with a discussion of limitations to mixing model applications. Through MixSIAR, we have consolidated the disparate array of mixing model tools into a single platform, diversified the set of available parameterizations, and provided developers a platform upon which to continue improving mixing model analyses in the future.

# INTRODUCTION

Mixing models, or models used to estimate the contribution of different sources to a mixture, are widely used in the natural sciences. Typically, these models require tracer data

that characterize the chemical or physical traits of both the sources and mixture—these traits are assumed to predictably transfer from sources to mixtures through a mixing process. In ecology, the majority of mixing model applications use stable isotope values as tracers in efforts to assess the contribution of prey (sources) to a consumer (mixture) diet, although other applications include animal movement, pollutant sourcing, cross-ecosystem nutrient transfer, and sediment erosion fingerprinting (*Bicknell et al., 2014*; *Bartov et al., 2012*; *Granek, Compton & Phillips, 2009*; *Blake et al., 2012*). In recent years, researchers have leveraged other tracers, such as fatty acid profile data to assess predator-prey relationships (*Neubauer & Jensen, 2015*; *Galloway et al., 2015*). Regardless of the tracers or mixing system considered, all mixing model applications are rooted in the same fundamental mixing equation:

$$Y_j = \sum_k p_k \mu_{jk}^s,$$

where the mixture tracer value, $Y_j$, for each of $j$ tracers is equal to the sum of the $k$ source tracer means, $\mu_{jk}^s$, multiplied by their proportional contribution to the mixture, $p_k$. This basic formulation assumes that (1) all sources contributing to the mixture are known and quantified, (2) tracers are conserved through the mixing process, (3) source mixture and tracer values are fixed (known and invariant), (4) the $p_k$ terms sum to unity, and (5) source tracer values differ. Given a mixing system with multiple tracers such that the number of sources is less than or equal to the number of tracers + 1, the $p_k$ terms in the set of $Y_j$ equations can be solved for analytically, given the unity constraint (*Schwarcz, 1991*; *Phillips, 2001*). In most natural mixing systems an analytical solution to the set of mixing equations is not possible without simplifying the mixing system or the data. In other words, in order to establish a solvable set of equations, researchers have traditionally reduced the number of sources through aggregation (*Ben-David, Flynn & Schell, 1997*; *Szepanski, Ben-David & Van Ballenberghe, 1999*). Additionally, because the analytic solution requires that the source and mixture data to be fixed (invariant), researchers used the source and mixture sample means and ignored uncertainty in these values (*Phillips, 2001* and references therein).

More recently, researchers have turned to more sophisticated mixing model formulations that provide probabilistic solutions to the mixing system that are not limited by the ratio of sources to tracers (i.e., under-determined systems), and that integrate the observed variability in source and mixture tracer data. The first of such models, IsoSource (*Phillips & Gregg, 2003*), provided distributions of feasible solutions to the mixing system based on a "tolerance" term; IsoSource iteratively identified unique solutions for the $p_k$ terms that resulted in $Y_j$ solutions falling near the true value of the mixture (typically defined by the mean of mixture data), where "near" was arbitrarily defined by the model user through the specification of tolerance. Subsequently, *Moore & Semmens (2008)* introduced a Bayesian mixing model formulation, MixSIR, that established a formal likelihood framework for estimating source contributions while accounting for variability in the source and mixture tracer data. An updated version of this modeling tool with a slightly different error parameterization, SIAR, continues to be broadly applied in

the ecological sciences and beyond (*Parnell et al., 2010*). Since 2008, Bayesian mixing models have rapidly evolved to account for hierarchical structure (*Semmens et al., 2009*), uncertainty in source data mean and variance terms (*Ward, Semmens & Schindler, 2010*), covariance in tracer values (*Hopkins & Ferguson, 2012*) and covariates within the mixing system (*Francis et al., 2011*). In short, Bayesian mixing models have developed into a flexible linear modeling framework, summarized by *Parnell et al. (2013)*.

In light of these analytic innovations, we have created an open-source R software package, MixSIAR, that unifies the existing set of mixing model parameterizations into a customizable tool that can meet the needs of most environmental scientists studying mixing systems. MixSIAR can be run as a graphical user interface or script, depending on the user's familiarity with R. Either version can be used to load data files and specify model options; then MixSIAR writes a custom JAGS (Just Another Gibbs Sampler, *Plummer, 2003*) model file, runs the model in JAGS, and produces diagnostics, posterior plots, and summary statistics. As with any sophisticated modeling tool, researchers should take care in establishing situation-specific applications of the tool based on the data in hand and the mixing system targeted for inference. At present, however, guidance on the parameterization and implementation of Bayesian mixing model analyses is lacking in the literature. As a consequence, many researchers are unsure of the correct application and interpretation of existing mixing model tools such as MixSIR (*Moore & Semmens, 2008*) and SIAR (*Parnell et al., 2010*).

In this paper we introduce and provide guidance on using MixSIAR for the application of Bayesian mixing models. Given early debate in the literature regarding appropriate error parameterizations (*Jackson et al., 2009*; *Semmens, Moore & Ward, 2009*), we begin by clarifying the underlying error structures for MixSIAR and provide recommendations for the use of specific error formulations based on the methods of data collection. The integration of prior information is a key advantage of Bayesian approaches to model fitting. However, since *Moore & Semmens (2008)*, few studies have implemented methods for generating prior distributions in mixing model formulations. We therefore provide a set of basic approaches to establishing prior distributions for the proportional contribution terms, and demonstrate how to incorporate informative priors in MixSIAR. Next, we provide guidance for source assignment in the mixing system (e.g., lumping or splitting source groupings). Arguably, the primary advantage of MixSIAR over previous mixing model software is the ability to incorporate covariate data to explain variability in the mixture proportions via fixed and random effects. As such, we provide guidance on applying covariate data within mixing models and illustrate this using MixSIAR in a case study on *American alligator* (*Alligator mississippiensis*) diet partitioning. Finally, we discuss limitations of mixing models and issues with under-determined systems. The complete set of MixSIAR equations with additional explanation is included as Article S1, and the MixSIAR code is available at https://github.com/brianstock/MixSIAR.

## Understanding MixSIAR error structures for mixture data

In most published results stemming from Bayesian mixing models, little if any detail is reported regarding the assumed error structure of the mixture data (literature review in

*Stock & Semmens, 2016b*). However, assumptions about variability, and the specific parameterizations used to characterize this variability, in the mixing system have been the focus of most of the innovations in mixing model tools in recent years (*Parnell et al., 2010, 2013; Ward, Semmens & Schindler, 2010; Hopkins & Ferguson, 2012; Stock & Semmens, 2016b*). The specific error formulation matters both because it relates to the assumptions regarding how the process of mixing occurs (e.g., how consumers feed on prey populations), and because the estimates of proportional source contributions can be affected (*Stock & Semmens, 2016b*). In this section, we discuss the suite of error parameterizations available in MixSIAR that account for variability in the tracer values of the mixture. Note that this section deals only with "residual" variability in the mixture tracer data after accounting for variability resulting from fixed or random effects (see Case study and Article S1 for how these effects interact with the error terms). For simplicity in the equations below, we ignore discrimination factors, concentration dependence and tracer covariance in our notation. Note, however, that MixSIAR accounts for each of these components, should an analyst specify a model appropriate to do so (see Article S1 for complete MixSIAR equations).

Researchers sometimes use "integrated" or "composite" sampling—pooling many subsamples into one sample that is then analyzed—to characterize the source means while keeping processing time and costs low, or because individual consumers or prey do not have enough biomass to analyze (*Hershey et al., 1993; Vander Zanden & Rasmussen, 1999; Grey, Jones & Sleep, 2001*). Thus, the most basic formulation for mixing models implemented in MixSIAR assumes that the $k$ source means for the $j$ tracers, $\mu_{jk}^s$, are fixed and invariant (but might be observed imperfectly; Fig. 1A). Under this assumption the mixture value for each tracer will also be an invariant weighted (by source proportions, $p_k$) combination of the source means. Observations of these means, however, are imperfect and thus the $i$ mixture data for tracer $j$, $Y_{ij}$, are assumed to follow the distribution,

$$Y_{ij} \sim N\left(\sum_k p_k \mu_{jk}^s, \sigma_j^2\right), \tag{1}$$

where $\sigma_j^2$ represents residual error variance, or the variability in observations associated with the mixture data points for the $j$th tracer. This error distribution is appropriate in situations where, for instance, each mixture data point was generated through the combination of many samples from the source population. For instance, if an analyst were interested in assessing the relative contributions of dissolved organic carbon and particulate organic matter to a filter feeder's diet, this model formulation would be appropriate since each source tracer datum comes from an integrated sample of the source tracer values (as opposed to tracer values of individual particles).

In contrast, for many mixing models applied to ecological systems, the tracers of individual source items (prey, e.g., individual deer) and mixtures (consumers; e.g., individual wolves) are analyzed separately, and the variability across source tracers is assumed to translate into consumer tracer variability—in other words, different wolves eat different deer, and their tracer values should differ accordingly (*Semmens et al., 2009*). Since the introduction of Bayesian stable isotope mixing models, nearly all published

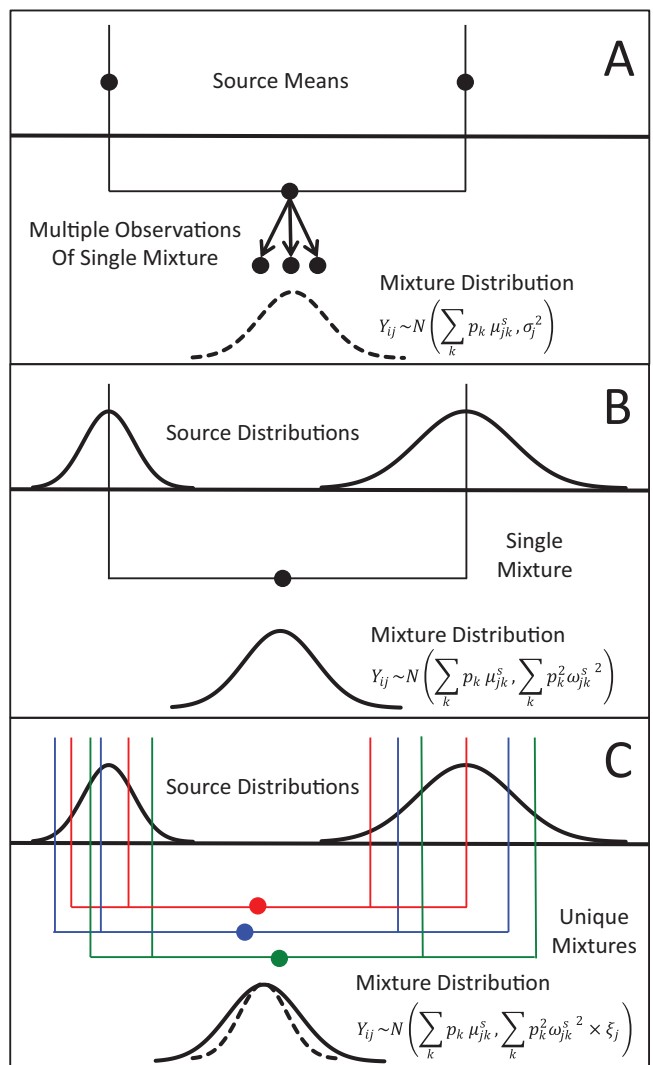

**Figure 1** **Representation of the three different methods MixSIAR uses for modeling variability in mixture data, assuming a two source ($k$), 1 tracer ($j$) scenario.** (A) In the "residual error only" for-mulation, the means of each source (upper black dots; typically estimated within the model based on source data) are additively combined, after weighting based on estimated proportional source con-tributions, in order to generate the expected mean value of the mixture signatures (Eq. 1). Actual mixture measurements deviate from this mean due to residual error, $\sigma_j^2$. (B) Given a single mixture data point, MixSIAR assumes this mixture value is drawn from a normal distribution defined by the same mean, with the variance generated by a weighted combination of source variances (Eq. 2). (C) In the "multiplicative error" formulation (Eq. 3), the model assumes the mixture data are generated from the process as in (B), but the variance of this distribution is modified by a multiplicative term, $\xi_j$, that allows the distribution to shrink (as would be expected if consumers are sampling multiple times from each source pool) or expand (as would be expected if the model is missing a non-negligible source, or processes such as isotopic routing introduce significant additional variability into the mixing system).

formulations have assumed that each mixture data point $i$ for tracer $j$ is derived from a normal distribution with the same mean as in Eq. (1), and, importantly, a variance similarly generated from a weighted combination of source variances, $\omega_{jk}^{s2}$:

$$Y_{ij} \sim N\left(\sum_k p_k \mu_{jk}^s, \sum_k p_k^2 \omega_{jk}^{s2}\right). \qquad (2)$$

In situations where there is covariance in tracers (typical of stable isotope studies), Eq. 2 can be modified to account for a weighted average of source covariance matrices (*Stock & Semmens, 2016b*).

MixSIAR uses this model formulation only in the special case where the analyst provides a single mixture value for each of the $j$ tracers considered. This formulation must be used in this special case because it is not possible to estimate a variance term, $\sigma_j^2$, from a single data point. In diet partitioning applications, the above formulation assumes that, for a given tracer $j$, a consumer $i$ takes one independent and identically distributed (IID) sample from each of $k$ sources and combines these samples in accordance with the proportional estimates $p_k$. In other words, each wolf eats exactly one deer, and thus incorporates the tracer value of only that deer. Because the prey-specific tracer values will be different for each consumer due to sampling error, the weighted combination of sampled source tracer values will also vary. We refer to this model of mixture variance as "process error" because it is derived from an assumption about the mixing process.

Recently, *Stock & Semmens (2016b)* modified the above formulation to include an additional multiplicative error term for each tracer considered, $\xi_j$ such that

$$Y_{ij} \sim N\left(\sum_k p_k \mu_{jk}^s, \sum_k p_k^2 \omega_{jk}^{s2} \times \xi_j\right). \qquad (3)$$

The intent of the $\xi_j$ term is to both add biological realism in the mixing equation, and to provide flexibility on the likelihood error structure such that mixing data not conforming to the mixing process assumed in the previous likelihood formulation can still be fit appropriately. As before, Eq. 3 can be modified to account for a weighted average of source covariance matrices (see Article S1). This model formulation is appropriate for most ecological mixing model applications (e.g., diet partitioning), with the exception of integrated sampling studies or studies with a single consumer sample, as outlined above. *Stock & Semmens (2016b)* showed that, compared to existing models (MixSIR, SIAR), Eq. 3 had lower error in $p_k$ point estimates and narrower 95% CI when the true mixture variance is low ($\xi_j < 1$).

When $\xi_j$ is less than 1, the variance in consumer tracer values shrinks, presumably due to the biological process of sampling each prey source multiple times from a distribution of tracer values (Fig. 1C). As the number of IID samples a consumer takes from a source population increases, the tracer value transferred from the source to the consumer will conform more and more closely to the mean source value. In other words, each wolf eats more than one deer, and thus each wolf incorporates a sample mean of deer tracer values, which becomes closer to the deer tracer mean as the number of deer sampled increases. Thus, $\xi_j$ indicates the amount of food a consumer integrates within a time frame determined by tissue turn-over; the methods for estimating this consumption rate are outlined in *Stock & Semmens (2016b)*. As the value of $\xi_j$ approaches zero,

 

an analyst can assume that the consumers are essentially "feeding at the mean" of the source populations.

Estimates of $\xi_j$ much greater than one indicate that the variability in transfer of tracer values from source to consumer is swamping the reduction in consumer variability expected when consumers integrate over multiple samples from prey populations. This could be due to factors such as isotopic routing (*Bearhop et al., 2002*), or important consumer population structure being absent from the model (e.g., most variability in wolf stable isotope values is explained by random effects of region and pack in *Semmens et al., 2009*). Alternatively, the mixing model could be missing a source or underestimating the source variances. In any case, values of $\xi_j$ much greater than one are an indication that the mixing system is not conforming to one or more of the basic assumptions of the mixing model, namely that tracers are not being consistently conserved through the mixing process, all mixtures are not identical (often not the case in biological systems), all mixtures do not have the same source proportions, or that the model is missing at least one source pool.

## CONSTRUCTING INFORMATIVE BAYESIAN PRIORS

### Priors for compositional data

The analysis of compositional data is not unique to mixing models. Examples of statistical models for compositional data are widespread in ecology (*Jackson, 1997*), fisheries (*Thorson, 2014*), as well as non-biological fields (*Aitchison, 1986*). The most common choice of prior on the estimated vector of proportions $p$ is the Dirichlet distribution; MixSIAR uses this distribution for estimates of source proportions. The Dirichlet is often referred to as a multivariate extension of the Beta distribution, and it is important to understand the Beta before transitioning to the Dirichlet. The Beta distribution has a convenient property that when both its shape parameters are 1, it is equivalent to a uniform distribution. In other words, if a model tries to estimate the relative contribution of a 2-component mixture, $p_1 \sim \beta(1, 1)$ is equivalent to $p_1 \sim Uniform(0, 1)$. Because the vector of proportions is constrained so that $\sum_{i=1}^{n=2} p_i = 1$, $p_2$ can be treated as the derived parameter $p_2 = 1-p_1$, and therefore does not require a prior. For the parameter of interest $p_1$, one way to describe the prior distribution is that the $\beta(1, 1)$ prior is uniform, and an equivalent description is that all possible combinations of $p_1$ and $p_2$ are equally likely *a priori*.

For mixtures with more than 2 components, MixSIAR uses the Dirichlet distribution to specify a prior on $p$. The hyperparameter of the Dirichlet distribution is a vector $\alpha$, whose length is the same as $p$. Like the Beta distribution, the only constraint on the elements of $\alpha$ is that they be positive (they may be discrete or continuous, and the elements of $\alpha$ don't have to be equal). A common choice of hyperparameters for a 3-component mixture is $\alpha = (1, 1, 1)$, which we refer to as the "uninformative"/generalist prior because (1) while every possible set of proportions has equal probability, the marginal prior likelihood of a given $p_k$ differs across values of $p_k$, and (2) its mean is $\left(\frac{1}{3}, \frac{1}{3}, \frac{1}{3}\right)$, corresponding to the assumption of a generalist diet (*McCarthy, 2007*). The first point is illustrated by Fig. 2, which shows that the marginal distributions of the

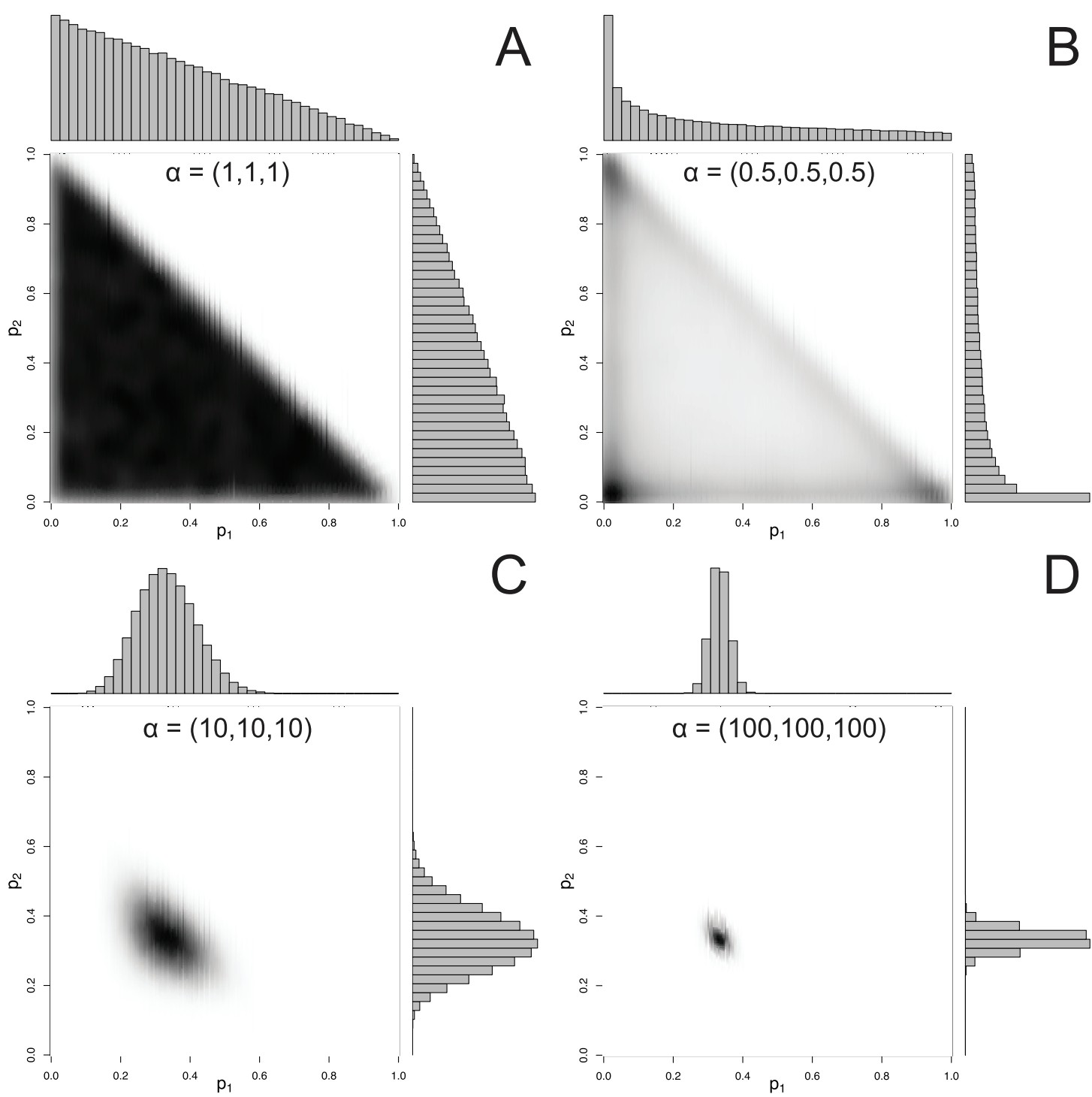

**Figure 2 Examples of joint and marginal distributions of $p_1$ and $p_2$ for a three-component Dirichlet distribution, across 4 sets of hyperparameters.** (A) $\alpha = 1$, (B) $\alpha = 0.5$, (C) $\alpha = 10$, and (D) $\alpha = 100$. All simulations were done with the "rdirichlet" function in the "compositions" library in R ( *Van Der Boogaart & Tolosana-Delgado, 2006*).

proportions are not uniform, instead favoring small values. Part of this confusion can be resolved by examining the joint pairwise distributions of $p$ (Fig. 2), which illustrates that using a hyperparameter of $\alpha = (1, 1, 1)$ implies that all combinations of $(p_1, p_2, p_3)$ are equally likely. Thus, this prior is noninformative on the simplex, but is non-uniform with respect to individual $p_k$ parameters. Other choices of a prior may be Jeffreys' prior, $\alpha = \left(\frac{1}{2}, \frac{1}{2}, \frac{1}{2}\right)$, or the more recently used logit-normal and extensions (*Parnell et al., 2013*). By default, MixSIAR uses the "uninformative"/generalist prior, where all $\alpha_k$ are set to 1.

## Constructing an informative prior

One of the benefits to conducting mixture models in a Bayesian framework is that information from other data sources can be included via informative prior distributions (*Moore & Semmens, 2008*). Once an informative prior for the proportional contribution of sources is established, MixSIAR can accept the prior as an input during the model specification process (for details and examples, see *Stock & Semmens, 2016a*). For diet studies, these other information sources are often fecal or stomach contents (*Moore & Semmens, 2008*; *Franco-Trecu et al., 2013*; *Hertz et al., 2017*), but can also include prey abundance or expert knowledge (*deVries et al., 2016*). As a simplified example from *Moore & Semmens (2008)*, suppose we wish to construct an informative prior for a 3-source mixing model of 10 rainbow trout diet using sampled stomach contents (30 eggs, 8 fish, 25 invertebrates). The sum of the Dirichlet hyperparameters roughly correspond to prior sample size, so one approach would be to construct a prior with $\alpha = (30, 8, 25)$, where each $\alpha_k$ corresponds to the source $k$ sample size from the stomach contents. A downside of this prior is that a sample size of 63 represents a very informative prior, with much of the parameter space given very little weight (Fig. 3). Keeping the relative contributions the same, the $\alpha_k$ can be rescaled to have the same mean, but different variance. One starting point is to scale the prior to have a total weight equal to the number of sources, $K$, which is the same weight as the "uninformative"/generalist prior:

$$\alpha_k = \frac{kn_k}{\sum n_k} \tag{4}$$

The prior constructed from Eq. 4 is shown in Fig. 3. Though this rescaling process of Dirichlet hyperparameters may seem arbitrary, it provides a powerful tool for incorporating additional information. Whether using this rescaled prior or not, we recommend that MixSIAR users always plot their chosen prior using the provided "plot_prior" function (Fig. 4).

Importantly, choosing a prior—including the "uninformative"/generalist prior—requires explicit consideration of how much weight the prior should have in any analysis. An additional consideration is the turnover time for different types of data. In our example of rainbow trout diet, stomach contents might represent a daily snapshot of prey consumption, whereas stable isotope and fatty acid values likely change on a much longer time scale (e.g., weeks to months). In such cases, we would want to downweight the prior's significance, since a prior constructed from daily information should only be loosely informative on the mixture proportions averaged over weeks to months.

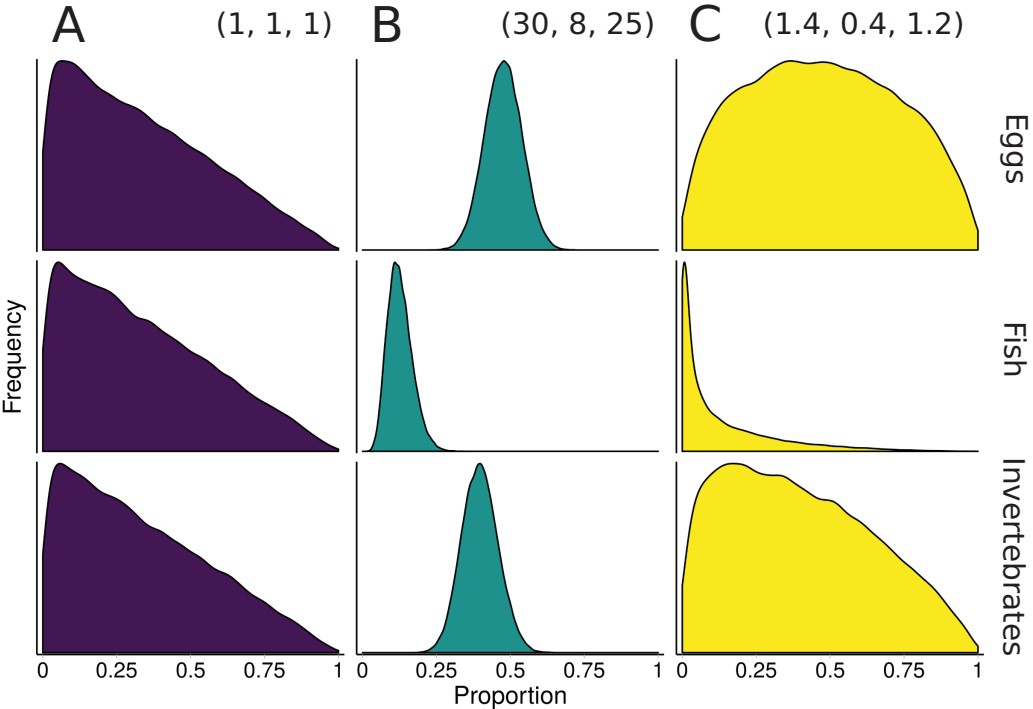

**Figure 3 Illustration of alternative priors for a mixing model of rainbow trout (consumers/mixture) diet comprised of three sources: eggs, fish, and invertebrates.** (A) The "uninformative"/generalist Dirichlet prior MixSIAR uses by default, $\alpha = (1, 1, 1)$. (B) A strongly informative prior with $\alpha = (30, 8, 25)$, where each $\alpha_k$ corresponds to the sample size of source $k$ from stomach contents. (C) A moderately informative prior with the same mean, but each $\alpha_k$ rescaled such that $\Sigma\alpha_k = 3$, the number of sources. Note that both informative priors have the same mean but differ in their "informativeness."

Exactly how much to downweight is unclear. However, this challenge lies within the broader issue of how to weight multiple data types, and we follow *Francis' (2011)* recommendation that users conduct a sensitivity analysis—fit the model using different informative priors (as well as the "uninformative"/generalist prior) and determine how sensitive the primary result is to the choice of prior (as in *deVries et al., 2016*).

## Priors for other model parameters

In addition to specifying prior distributions on proportional contributions, MixSIAR requires priors on variance parameters (*Parnell et al., 2013*). Because mixing models ultimately are a class of linear models, MixSIAR uses the same weakly informative prior distributions for variances that are widely used in other fields (*Gelman et al., 2014*). For specific prior formulations associated with residual error, multiplicative error, and variance associated with random effects, we refer the reader to the full set of MixSIAR equations (Article S1). Note, however, that because MixSIAR generates a model file in the JAGS language (Just Another Gibbs Sampler, *Plummer, 2003*) during each model run, the analyst can access the complete set of prior specifications associated with the model run. Moreover, the model file can be modified and used in a separate model run

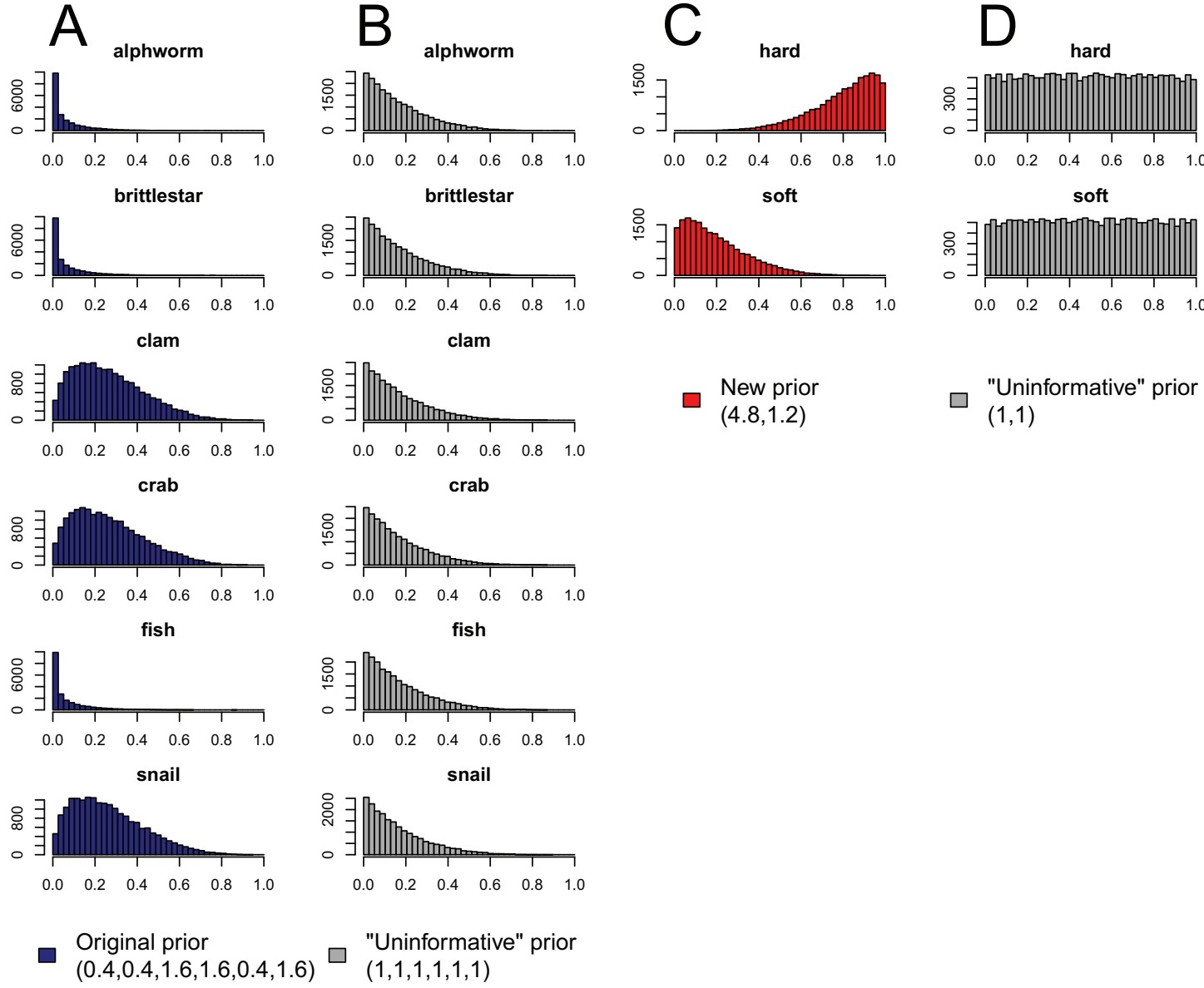

**Figure 4 Effect of aggregating sources *a posteriori* on priors in mixing models, produced by the "combine_sources" function in MixSIAR as a warning to the user.** (A) the original, unaggregated prior on six sources from the mantis shrimp example (dark blue); (B) the "uninformative"/ generalist prior on six sources (grey); (C) the prior resulting from aggregating the six-source prior in dark blue into two sources (hard-shelled = clam + crab + snail, soft-bodied = alphworm + brittlestar + fish, red); and (D) the prior resulting from aggregating the six-source "uninformative"/ generalist prior into the same two sources (grey).

outside of MixSIAR, should the analyst care to evaluate the sensitivity of model outputs to changes in prior specification.

In some cases, an analyst may wish to incorporate discrete or continuous covariates to explain differences between individual tracer values (*Francis et al., 2011*; *Ogle, Tucker & Cable, 2014*; "Incorporating covariates via fixed and random effects" section to follow). Ecological examples of these types of covariates may include environmental variables

(habitat, temperature) or variables specific to individuals (sex, age, size). Like simple linear regression, including covariates introduces new parameters to be estimated (intercept, slope), but because MixSIAR includes these covariates in transformed compositional space (isometric log ratio, ILR; *Aitchison, 1986*), their prior specification is not straightforward. MixSIAR uses diffuse normal priors in transform space, which are sufficient to establish priors that yield parameter estimates that are essentially informed only by the data (*Gelman et al., 2014*; *McElreath, 2016*). Analysts who wish to create informative priors in transform space should proceed with caution, because they can have counterintuitive effects when transformed back to proportion space. A future improvement to MixSIAR would be to allow users to run models without data to understand what the joint prior entails for the marginal proportions.

## INCORPORATING SOURCE DATA INTO MIXING MODELS

Early versions of Bayesian mixing models treated the estimates of source-specific tracer means and variance as fixed (user specified), and thus only used raw mixture data in calculating the likelihood of source proportions (*Moore & Semmens, 2008*; *Parnell et al., 2010*). In so doing, the uncertainty in the estimates of source means and variances, typically derived from source isotope data, was ignored. However, *Ward, Semmens & Schindler (2010)* introduced what they termed a "fully Bayesian" model that accounts for estimation uncertainty in source-specific tracer means and variances, and thus treats both the mixture and source information as data within the model framework. More recently *Hopkins & Ferguson (2012)* incorporated multivariate normality into estimates of source-specific covariance matrices. This multivariate normality accounts for the fact that tracer values often co-vary, particularly for stable isotope studies.

MixSIAR provides the analyst with each of these three options for including source data, because each can be appropriate in different circumstances (Article S1). In order of preference—but also model complexity—analysts can provide: (1) raw tracer data for each source, or (2) source tracer value summary statistics (mean, variance and sample size). In both cases, MixSIAR fits a fully Bayesian model by estimating the "true" source means and variances for each tracer (*Ward, Semmens & Schindler, 2010*; *Parnell et al., 2013*). However, in the case where summary statistics are provided, the tracers are assumed to be independent, since it is not possible to generate estimates of tracer covariance from the summary statistics. Where raw source data are provided, MixSIAR assumes multivariate normality and estimates the variance covariance matrix associated with the tracers for each source (*Hopkins & Ferguson, 2012*). This normality assumption does not hold for compositional tracer (e.g., fatty acid profile) data, and therefore we advise users with such data to use the second option above (see Article S1 and S2). Alternatively, analysts can use other software packages specifically designed to accommodate fatty acid data (QFASA, *Iverson et al., 2004*; fastinR, *Neubauer & Jensen, 2015*). The third, and final, option is to specify fixed (known) source means and variances, which approximates MixSIR (*Moore & Semmens, 2008*) and SIAR (*Parnell et al., 2010*). Analysts can accomplish this in MixSIAR by providing summary statistics (mean and

variance) with an arbitrarily large sample size (~10,000). This approach essentially fixes the estimated source means and variances at the values provided.

## Combining sources

No amount of increased sophistication in mixing model methods can overcome the problem of poorly specified mixing systems. If, for instance, an analyst specifies a mixing model with >7 sources contributing to a mixture based on two tracers (e.g., $\delta^{13}C$, $\delta^{15}N$), it is unlikely the model products will be precise or interpretable. The source data (number of sources and their sample sizes, means, and variances relative to mixture data) have a large influence on the estimated proportions. As such, including several largely extraneous sources with few mixture data points will divert $p_k$ from the truly important sources (as $\sum p_k = 1$). We note, however, that there are ways to constrain the $p_k$ such that models converge—two methods discussed in other sections are informative priors and including covariates on the $p_k$ as fixed or random effects. Nonetheless, MixSIAR can estimate posterior distributions of source proportions regardless of how under-determined the mixing system is (e.g., many more sources than tracers). This under-determination, together with the variability in source and mixture isotopic values, often results in quite diffuse probability distributions for many of the proportional contribution estimates, limiting the interpretability of the results (*Phillips et al., 2014*). Reducing the number of sources by combining several of them together may improve model inference. Either *a priori* or *a posteriori* aggregation (*Phillips, Newsome & Gregg, 2005*) may be used with MixSIAR (see "combine_sources" function for *a posteriori* aggregation).

The *a priori* approach typically involves pre-processing the input data by conducting frequentist tests for equality of means of sources and subsequently combining sources without significant differences before running a mixing model (*Ben-David, Flynn & Schell, 1997*). If tracer data are approximately normally distributed, a Hotelling's $T^2$ test can be used to evaluate whether sources are not different from each other, given multivariate data (multiple tracers; *Welch & Parsons, 1993*). If tracers are not normally distributed, a $K$ nearest-neighbor randomization test can be used to assess differences in sources (*Rosing, Ben-David & Barry, 1998*). Note that in both cases, a Bonferroni-type correction is typically used when multiple source comparisons are made. Regardless of the test used, if sources appear similar, their data can be aggregated. In general, mixing model outputs will be more interpretable if the sources combined have a logical connection (e.g., same trophic guild, taxon, etc.) so that the aggregated source has some biological meaning, rather than a disparate set of unrelated sources that happen to have similar isotopic values, although this is not an absolute requirement.

Using a frequentist approach (e.g., Hotelling's $T^2$ test) to decide on whether sources should be combined *a priori* often presents problems. The amount of data available for each source directly influences the equality of the means tests; the power to reject a null hypothesis of no mean difference between tracer values of sources is thus related to the amount of tracer data, and is not exclusively a function of the mixing system. Furthermore, in situations when many tracers are available (e.g., fatty acids

as tracers; *Galloway et al., 2015*) there is a high probability that at least some equality of mean tests will fail (reject the null hypothesis) even if the sources are, in reality, identical. Finally, when only the mean, variance and sample size of each source is available (rather than raw data), there is no easy test for equality of the means and methods for aggregating sources are not apparent.

Using the *a posteriori* procedure, the analyst uses the full set of sources to generate posterior distributions of proportional source contributions, and then post-processes the results to combine several sources together. For each posterior draw, the new combined source proportion is simply the sum of the proportions of the original sources. Thus, we obtain a posterior distribution for the new combined source proportion that accounts for correlation between the original source proportions. This new posterior distribution may then be analyzed as before. Importantly, this approach does not require that the tracer values of the combined sources are similar; thus, an analyst is free to combine sources based on functional similarities in the mixing system, regardless of tracer similarity.

Like the *a priori* approach, combining posteriors from multiple sources as a means of source aggregation is not without issues. One caveat is that each additional source included in the mixing model increases the number of parameters to be estimated, particularly when the model includes random effects. We could easily imagine that a mixing model with 20 sources and random effects may take days to run successfully, and may not converge at all. In models with many more sources than tracers, the source proportions are more likely to be confounded, and therefore highly negatively correlated. In such cases, it is less likely the model will converge. Another potential issue with the *a posteriori* approach is that the combination of multiple diet proportions estimated with an "uninformative"/generalist Dirichlet prior (each source given equal prior weight) also combines the prior weight for these sources. For instance, given an "uninformative"/ generalist Dirichlet prior, the act of aggregating two source posteriors results in a combined source posterior that reflect an aggregated prior with twice the weight of the remaining non-aggregated source priors. As such, the more sources that are combined into an aggregate source group *a posteriori*, the more strongly the prior will be weighted towards increased proportional contributions of this aggregate source to the consumer diet. MixSIAR alerts users to this issue by plotting the aggregated prior when combining sources using the "combine_sources" function (Fig. 4). This is not an issue, however, when the same number of sources are combined into new groupings (e.g., *deVries et al., 2016*, where six sources were combined into two groups of three). In general, combining sources *a posteriori* can lead to lower variance in diet proportion estimates, particularly when the posteriors for the combined sources show strong negative correlation (*Semmens et al., 2013*). For most situations, we prefer the *a posteriori* approach to source aggregation, provided the analyst is aware of the cautions mentioned above.

These *a priori* and *a posteriori* approaches to combining sources may be accomplished by simple pre-processing of MixSIAR input data sets and post-processing of MixSIAR output using the "combine_sources" function, respectively. *Ward et al. (2011)* outlined a Bayesian approach that probabilistically identifies source groupings and generates weighted posterior probabilities associated with various combinations of sources.

However, their method requires specialized Markov chain Monte Carlo (MCMC) sampling, and is computationally impractical for complicated mixing systems. We expect that future refinements to the modeling approach they outlined will yield more robust techniques for treating source combinations as parameters to be estimated, rather than fixed *a priori* or *a posteriori*.

## INCORPORATING COVARIATES VIA FIXED AND RANDOM EFFECTS

In many cases, covariate data (also called explanatory or independent variables) are available for incorporation into a Bayesian mixing model to answer important questions about the mixture (*Francis et al., 2011*; *Ogle, Tucker & Cable, 2014*). Neglecting to include covariates that are relevant to the mixture proportions can lead to pseudoreplication, since the model assumes all mixtures are IID (*Hurlbert, 1984*). Some examples from diet partitioning applications include:

1. Consumers (mixtures) are of different sexes and an analyst has interest in whether the dietary proportions differ between sexes (fixed categorical effect).
2. An analyst has additional numerical measures on the consumers such as weight, length, etc., and would like to see whether the dietary proportions are affected by this value (fixed continuous effect).
3. An analyst has samples of consumers or sources in different regions. It is likely that the consumers' dietary proportions are similar between regions so it makes sense that the estimates should 'borrow strength' between the groups (random effect).

In each case it is possible to run a traditional mixing model separately for each sex, region, time point, etc. However, this process can be time-consuming and will often lead to inefficient inference with greater uncertainties in the dietary proportions for three main reasons. First, there will be no direct estimate of the effect size between groups. Second, additional residual error terms will be fit (a residual error term for each level of the fixed/random effect, instead of one error term shared across levels). Third, there is no way to "borrow strength" between groups, since each set of dietary proportions must be estimated independently. The solution lies in adding the extra information as covariates through the dietary proportions in the mixing model directly. To illustrate the application of fixed and random effects using MixSIAR software we describe a case study on *Alligator mississippiensis* diet partitioning, which executes multiple model formulations and evaluates their relative support using information criteria (*Nifong, Layman & Silliman, 2015*; for data and R code see Data S1).

A common question is how to choose whether to use fixed or random effects. We recognize that the terms "fixed" and "random" effects are unclear (*Gelman, 2005*), and in Gelman's "constant" versus "varying" terminology, both fixed and random effects in MixSIAR are varying (different for each factor level). Nonetheless, *Gelman (2005)* recommends using random effects (as defined in MixSIAR, Article S1) when possible, since borrowing strength between groups is a desirable property, and always allows for the model

to choose large random effect standard deviations that will yield nearly equivalent estimates to those resulting from fixed effects structure when the analyst has reasonably informative isotopic data. The random effects model draws offsets from a shared distribution, which is appropriate if the factor levels are related, as they often are in biological systems. The random effects model also allows inference on the relative importance of multiple factors through variance partitioning. For example, *Semmens et al. (2009)* showed that for British Columbia wolves, $\gamma^2_{Region} > \gamma^2_{Pack} > \gamma^2_{Indivual}$, which means that Region explained most variance in wolf diet, followed by Pack and Individual. However, when the number of groups is small (<5) there can be difficulties in estimating the random effect standard deviations, and fixed effects should always be used when a factor has only two groups.

## Technical details

For covariates to be included, the model must allow for dietary proportions to be specified per individual, e.g., the mixture likelihood must be of a form similar to:

$$Y_{ij} \sim N\left(\sum_k p_{ik}\mu^s_{jk}, \sum_k p^2_{ik}\omega^2_{jk} * \xi_j\right).$$

Where $p_{ik}$ is now the dietary proportion for source $k$ on individual $i$.

Regardless of which fixed or random effects are used, MixSIAR establishes a base set of diet proportions $p$ using a Dirichlet prior that can be modified with prior information. Once specified, these proportions are isometric log-ratio (ILR) transformed into ILR-space parameters, $\beta_0$ (*Parnell et al., 2013*). This transformation maps a composition in the $k$-part Aitchison-simplex isometrically to a $k$-1 dimensional Euclidean vector. Each of the $\beta_0$ transformed components are normally distributed and independent of each other and can thus be broached by standard multivariate analysis methods.

Once transformed, these $\beta_0$ terms can be modified through the incorporation of covariates, and then subsequently back-transformed into individual-specific vectors of diet proportions $p_i$. For instance, for a simple fixed effects structure like that described in example 1 above, we have:

$$p_i = \text{inverse.ILR}(\beta_0 + \beta_1 \text{Sex}_i).$$

The parameters in the vector $\beta_1$ cumulatively represent the change in dietary proportions for the difference between female and male. In this instance, the categorical fixed effect $\text{Sex}_i$ is coded so that male = 1 and female = 0 (or vice versa).

If the covariate is continuous, as in example 2, the structure changes only very slightly:

$$p_i = \text{inverse.ILR}(\beta_0 + \beta_1 \text{Weight}_i).$$

Now the parameters in the vector $\beta_1$ represent the change in dietary proportions according to a unit increase in the weight of the consumer.

Covariates are included as random effects in a similar manner. For example 3 given above, we might have:

$$p_i = \text{inverse.ILR}\left(\beta_0 + \beta_{\text{Region}(i)}\right)$$

where each of the $k-1$ random effect terms in the vector $\boldsymbol{\beta}_{\mathbf{Region}(i)}$, have an extra constraint: $\boldsymbol{\beta}_{\mathrm{Region}(i),k} \sim N\left(0, \gamma^2_{\mathrm{Region}}\right)$. This constraint allows the model to borrow strength between groups. If $\gamma^2_{\mathrm{Region}}$ is small, then the groups are similar and the dietary proportions will not change much between regions. If $\gamma^2_{\mathrm{Region}}$ is large however, the regions will be very different and this will be reflected in the dietary proportions. If multiple random effects are included in the model, the differences between $\gamma^2$ terms for each covariate illustrate their relative importance to the consumer diet (as in *Semmens et al., 2009*, where $\gamma^2_{\mathrm{Region}} > \gamma^2_{\mathrm{Pack}} > \gamma^2_{\mathrm{Individual}}$, indicating that Region explained more of the diet variability than Pack or Individual).

Since there is no one-to-one relation between the original parts and the transformed variables (i.e., each $\beta_k$ acts on all $p_k$ terms simultaneously), interpretation of model findings after back-transforming is prudent. MixSIAR therefore provides summary output statistics and preserves posterior draws on the back-transformed proportions for fixed categorical and random effects. In the case of continuous fixed effects (see below), MixSIAR generates a plot of the fitted line in the untransformed proportion space that spans the range of the provided covariate data. For the full set of MixSIAR equations and additional explanation, see Article S1.

## Case study: *Alligator mississippiensis* diet partitioning

This case study highlights the main advantage of MixSIAR over previous mixing model software—the ability to include fixed and random effects as covariates explaining variability in mixture proportions and calculate relative support for multiple models via information criteria. *Nifong, Layman & Silliman (2015)* analyzed stomach contents and stable isotopes to investigate cross-ecosystem (freshwater vs. marine) resource use by the American alligator (*Alligator mississippiensis*), and how this varied with ontogeny (total length), sex, and between individuals. They used 2-source (marine, freshwater), 2-tracer ($\delta^{13}$C, $\delta^{15}$N) mixing models and posed three questions:

Q1. What is $p_{marine}$ vs. $p_{freshwater}$?

Q2. How does $p_{marine}$ vary with the covariates Length, Sex, and Individual?

Q3. How variable are individuals' diets relative to group-level variability?

*Nifong, Layman & Silliman (2015)* grouped the consumers into eight subpopulations (all combinations of Sex: Size Class, where Sex $\in$ {male, female} and Size Class $\in$ {small juvenile, large juvenile, subadult, adult}) and ran separate mixing models for each using SIAR (*Parnell et al., 2010*). To calculate $p_{marine}$ estimates for the overall population, they also ran a mixing model with all consumers. In addition to inadequately addressing Q3 on individual diet variability, this approach is likely inefficient, as it fits nine residual error terms for each tracer and does not capitalize on the fact that diets of different-sex and different-sized alligators are probably related. We propose that a more natural, statistically efficient approach is to fit several models with fixed and random effects as covariates, and then evaluate the relative support for each model using information criteria (see "compare_models" function in MixSIAR).

**Table 1 Comparison of mixing models fit using MixSIAR on the alligator diet partitioning data from *Nifong, Layman & Silliman (2015)*.**

| Model | LOOic | SE (LOOic) | dLOOic | SE (dLOOic) | Weight | $\xi_C$ | $\xi_N$ |
|---|---|---|---|---|---|---|---|
| Length | 820.8 | 31.4 | 0 | – | 0.789 | 5.3 | 1.0 |
| Length + Sex | 823.6 | 31.4 | 2.8 | 2.1 | 0.195 | 5.2 | 1.0 |
| Size class | 829.5 | 31.6 | 8.7 | 11.7 | 0.010 | 5.4 | 1.1 |
| Size class + Sex | 831.4 | 31.5 | 10.6 | 12.1 | 0.004 | 5.3 | 1.1 |
| Size class: Sex | 832.9 | 29.8 | 12.1 | 13.6 | 0.002 | 4.9 | 1.1 |
| Habitat | 890.7 | 28.7 | 69.9 | 43.4 | 0 | 6.4 | 1.5 |
| Sex | 973.8 | 17.7 | 153.0 | 30.1 | 0 | 8.4 | 2.2 |
| – | 977.0 | 16.7 | 156.2 | 31.5 | 0 | 8.4 | 2.2 |

Notes:
dLOOic is the difference in LOOic between each model and the model with lowest LOOic. The "Length" model had the lowest LOOic and received 79% of the Akaike weight, indicating a 79% probability it is the best model. The "Length + Sex" model cannot be ruled out (20% weight). Note that as variability in the mixture data is better explained by covariates, the estimates of $\xi_j$ decrease.

We used MixSIAR to fit eight mixing models with different covariate structures (Table 1; Data S1). Since each model is fit to the same data ($\delta^{13}C$ and $\delta^{15}N$ values for each of 181 alligators), we can compare the models using information criteria. Deviance information criterion (DIC) is a commonly-used generalization of Akaike information criterion (AIC) for Bayesian model selection which estimates out-of-sample predictive accuracy using within-sample fits. DIC, however, has several undesirable qualities (e.g., can produce negative estimates of the effective number of parameters, is not defined for singular models, and is not invariant to model parameterization; *Vehtari, Gelman & Gabry, 2017*). Therefore, MixSIAR implements the widely applicable information criterion (WAIC) and approximate leave-one-out cross-validation (LOO), both of which are more robust to the concerns associated with DIC (*Vehtari, Gelman & Gabry, 2017*). For a set of candidate models fit to the same mixture data, we can calculate the relative support for each model using LOO and Akaike weights, which are estimates of the probability that each model will make the best predictions on new data (*Burnham & Anderson, 2002*; *McElreath, 2016*).

We found that the models with Length as a continuous fixed effect are heavily preferred over the models that break length into four size classes (combined weight of "Length" and 'Length + Sex' = 99%, Table 1). There is little evidence for including sex in addition to length or size class, although it cannot be ruled out (adding sex increases LOO in both cases, but "Length + Sex" still receives 20% weight, Table 1). While the original analysis by *Nifong, Layman & Silliman (2015)* predicts $p_{marine}$ as a function of subpopulation membership, the "Length" model predicts $p_{marine}$ as a function of length (Fig. 5). Under the "Size class:Sex" model of *Nifong, Layman & Silliman (2015)*, the $p_{marine}$ estimate for adult males is 0.76 (median, 95% CI [0.68–0.84]), while the "Length" model estimate of $p_{marine}$ for the largest individual, a 315.5 cm adult male, is 0.96 (median, 95% CI [0.91–0.99]). Although *Nifong, Layman & Silliman (2015)* clearly document an ontogenetic shift in alligator resource use, the data support the conclusion that this shift likely occurs as a continuous function of body size, instead of in discrete stages.

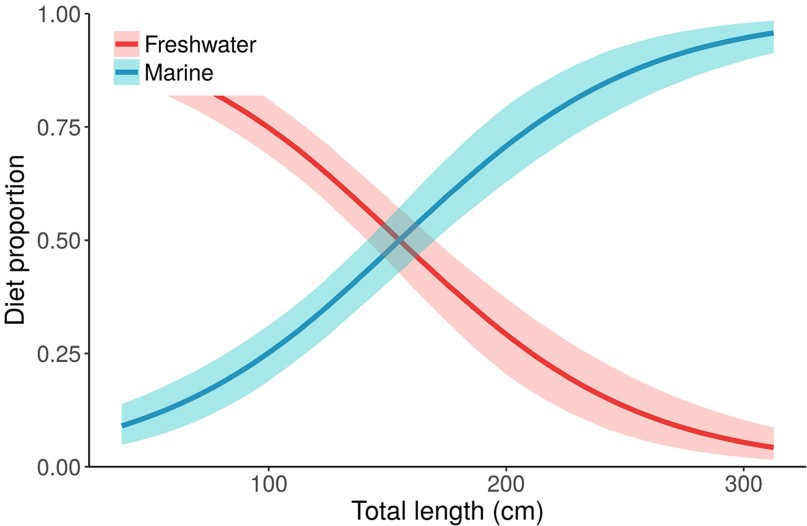

**Figure 5 Posterior distributions for alligator diet proportions as a function of length from the best performing model, "Length."** Small/young alligators depend upon freshwater prey and shift to a marine-based diet as they increase in size. Lines depict posterior medians, and shading displays the 90% credible intervals. The "Length" model estimate of $p_{\text{marine}}$ (blue curve) for the largest individual, a 315.5 cm adult male, is 0.96 (median, 95% CI [0.91–0.99]). Estimates of $p_{\text{marine}}$ for the smallest (37.7 cm) and median-sized (116.9 cm) alligators are 0.09 (0.04–0.15) and 0.32 (0.24–0.39), respectively.

This case study also highlights the interaction between covariates and the multiplicative error term, $\xi_j$. As covariates are included that increasingly explain the observed variability in alligator isotope values, the estimates of $\xi_j$ shrink ($\xi_C$ decreases from 8.4 to 5.2, $\xi_N$ decreases from 2.2 to 1.0; Table 1). The $\xi_N$ estimate from the "Length" model (1.0) is about what we expect given the assumptions about how predators sample prey. The $\xi_C$ estimate (5.2) is very high, however, indicating that there remains an important process that is unaccounted for in the model. There are several possible explanations (see section on "Understanding MixSIAR error structures for mixture data"), with one being that individuals' diets likely differ based on other processes than sex or length—all models in Table 1 assume that individuals of the same sex, length, or size class share the same diet proportions. We can, however, relax this assumption by including Individual as a random effect in addition to Length (or other covariates). Then the diet proportion for the $i$th individual becomes:

$$p_i = \text{inverse.ILR}(\beta_0 + \beta_1 \text{Length}_i + \beta_{\text{ind}}),$$

$$\beta_1 \sim N(0, 1000),$$

$$\beta_{\text{ind}} \sim N\left(0, \sigma_{\text{ind}}^2\right),$$

$$\sigma_{\text{ind}}^2 \sim U(0, 20).$$

This "Length + Individual" model allows $p_{\text{marine}}$ for individual alligators to vary around the expectation based on Length (Fig. 6).

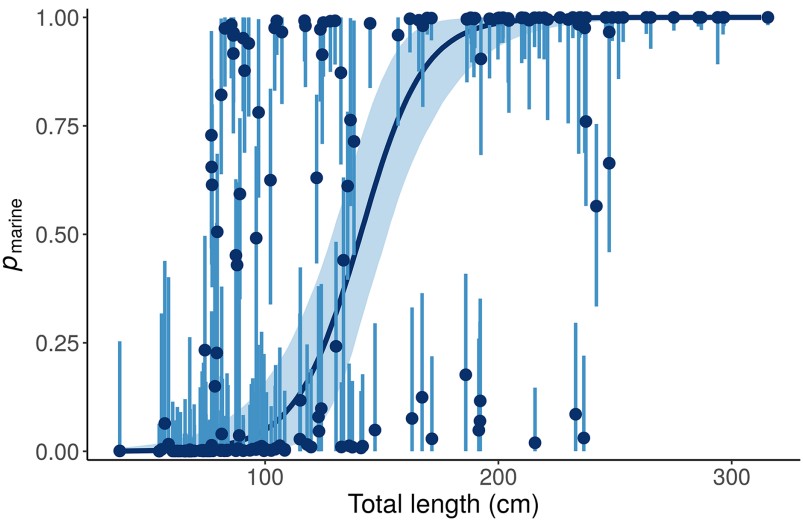

**Figure 6** **Posterior distributions for the marine proportion, $p_{marine}$, of alligator diet as a function of length from the "Length + Individual" model.** Whereas the "Length" model estimates one diet for all alligators of a given length, the "Length + Individual" model allows $p_{marine}$ for individual alligators to vary around the expectation based on Length. For most alligators around 100 cm total length, the $p_{marine}$ is very low, but for some it is above 80%. Likewise, the model estimates that most large (>200 cm) alligators' diets are dominated (>95%) by marine prey, but $p_{marine}$ for three large individuals is less than 10%. Dark blue line and points indicate posterior medians, light lines and shading show 90% credible intervals.

Like many ecologists, *Nifong, Layman & Silliman (2015)* were interested in how variable individuals' diets are, relative to group-level variability (Q3). They calculated the specialization index (ε) of *Newsome et al. (2012)* for their overall population model, 0.26 ± 0.05, concluded that alligators are mostly generalists, and "the diet of the majority of individuals is expected to be comprised of similar proportions of freshwater and marine prey." The proper interpretation, however, is clearer with the best performing model ("Length")—the specialization index of an alligator of *average length* is low, but small and large alligators are highly specialized (Fig. 7). Additionally, since the "Length + Individual" model estimates individuals' diet proportions, we can plot the distribution of $\varepsilon_{ind}$ and see directly that most alligators are specialists (ε > 0.8, Fig. 8). *Nifong, Layman & Silliman (2015)* performed a well-designed study, and their main conclusions are robust—we only reanalyze their data here to highlight advantages of MixSIAR over other mixing model software.

## LIMITATIONS OF BAYESIAN MIXING MODELS

Like any statistical model, inference from mixing models is only as good as the data being used. In some situations, data may not be informative—these situations may arise when models are misspecified or data are limited (i.e., there is a mismatch between the model structure and data structure). Such situations may be difficult to diagnose, and we encourage mixing model users to reach out to other users and contributors (https://github.com/brianstock/MixSIAR/issues). Some misspecifications are simple to fix, while other times they require a detailed examination of the likelihood or posterior

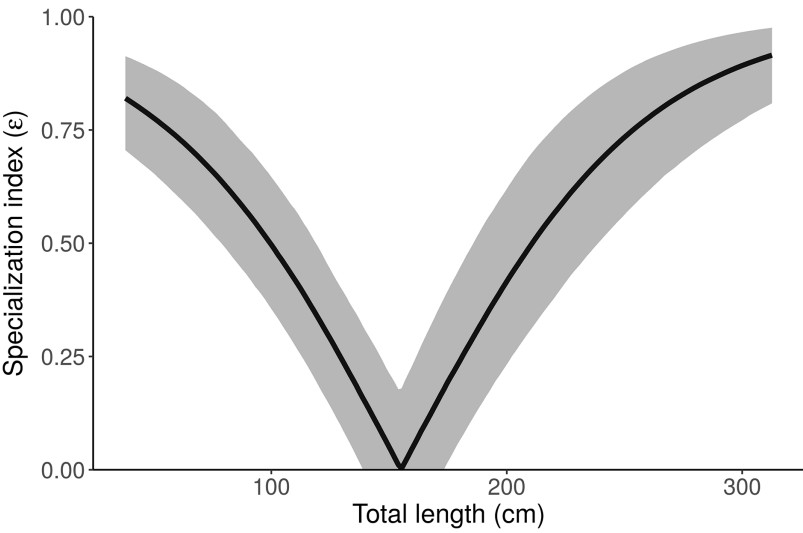

**Figure 7 Posterior distribution of the specialization index (ε) as a function of length from the "Length" model.** Small and large alligators are highly specialized (on freshwater and marine prey, respectively), whereas average-length alligators have low specialization index (i.e., are consuming both freshwater and marine prey). Specialization index is calculated using Eq. 5 in *Newsome et al. (2012)* from individual MCMC draws of $p_{freshwater}$ and $p_{marine}$ as a function of length. The line depicts the posterior median and shading displays the 95% credible interval.

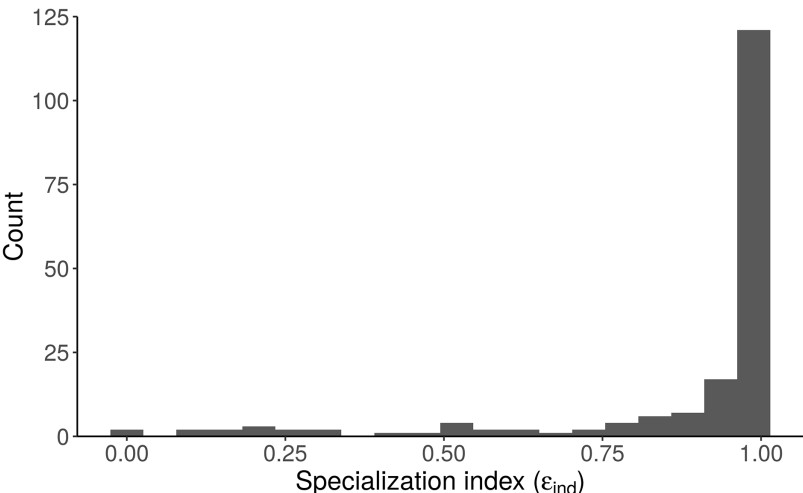

**Figure 8 Distribution of the specialization index calculated for each individual (ε$_{ind}$, *n* = 181) from the "Length + Individual" model estimates of individuals' diet proportions (posterior median of $p_{ind}$).** The model estimates that most alligators sampled by *Nifong, Layman & Silliman (2015)* are specialists (ε > 0.8).

distributions (which may appear flat with respect to the parameter of interest). Similar situations arise in all statistical models—for example fitting a regression model to a constant response $Y = (3,3,3, \dots )$ returns an estimate that is a perfect fit to the data, but does not produce standard errors or test-statistics (the response is assumed to be normally distributed, but the variance of $Y = 0$). Several recent papers have illustrated some of these same points with respect to mixing models, and we detail those here.

As a first limitation, *Bond & Diamond (2011)* illustrated that recently developed mixing models are sensitive to the choice of discrimination factors (systematic changes in the tracer values through the mixing process). This issue arises because the discrimination factors and estimated source contributions are not completely identifiable. In other words, these parameters are difficult to estimate simultaneously, and one or the other is generally fixed (in food web studies, the discrimination factor is typically specified as fixed *a priori*). At present, MixSIAR does not provide the option to estimate discrimination from user-provided data, although such functionality could easily be added; we anticipate adding this functionality into a future software release.

A second limitation of mixing models is that systems may be underdetermined (as discussed in Introduction). *Phillips & Gregg (2003)* demonstrated several examples of this problem for the 2-tracer scenario, but the issue of underdetermined problems generally arises when the number of sources exceeds the number of tracers plus one. In such instances, posterior estimates of source contributions can be broad and multi-modal, owing to the fact that multiple, often disparate, solutions to the underlying mixing equations exist. *Fry (2013)* proposed a graphical approach to separate data-supported aspects of solutions from any assumed aspects of solutions method. Essentially, this approach is a post hoc means of evaluating model performance, and can easily be applied to the products of any mixing model (including the products of a MixSIAR model run).

The influence of the Dirichlet prior on the source proportions is a separate, but related, issue—the prior becomes more influential with more sources. Contrary to how this discussion has been framed previously (*Brett, 2014*; *Galloway et al., 2014*, *2015*), the influence of the prior is not simply a matter of underdetermined-ness of the system, and therefore is not entirely avoided by increasing the number of tracers above the number of sources plus one (so that the system is not underdetermined; i.e., a model with 15 fatty acids and 12 sources is not underdetermined but still has this problem, Fig. 9). The influence of the Dirichlet prior also increases with fewer data points, greater source data variance, and less separation between sources. *Brett (2014)* described the interaction between these three factors (which determine the shape of the mixing polygon) and the prior as a bias of mixing models. This phenomenon may be better described as weakly informative data, but we agree that approaches like *Brett (2014)*'s surface area metric may be useful in recognizing *a priori* when these situations may arise. As such, we have incorporated Brett's surface area metric as a diagnostic output in MixSIAR ("calc_area" function). However, work still needs to be done to generalize this metric to situations with any number of tracers and sources.

## CONCLUSION

Analysts applying modern mixing model software typically must navigate a challenging array of model choices, from source groupings to covariate data, to error parameterization. In the past, those analysts not capable of developing their own models have been faced with the choice between different software packages, each with differing statistical model structures and assumptions. Through the creation of MixSIAR,

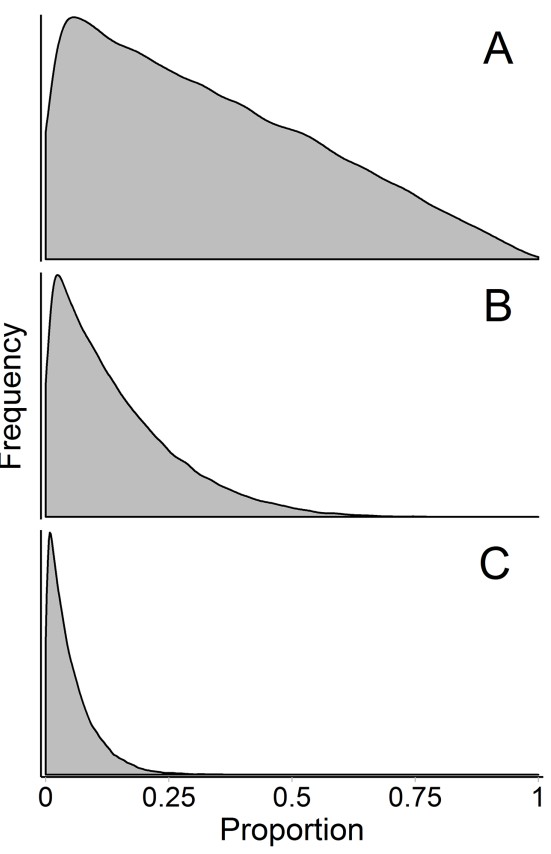

**Figure 9 Marginal source proportion distributions for the "uninformative"/generalist prior with increasing number of sources.** While the "uninformative"/generalist prior remains uninformative on the simplex in all cases, as the number of sources, $K$, increases from (A) $K = 3$, to (B) $K = 7$, and (C) $K = 20$, the Dirichlet prior becomes more informative on the marginal source proportions. For this reason, analysts should only include more than ~7 sources with extreme caution, even if the mixing system is not underdetermined. All simulations were done with the "rdirichlet" function in the 'compositions' library in R (*Van Der Boogaart & Tolosana-Delgado, 2006*).

we have incorporated the disparate suite of mixing model advances into a single tool with the flexibility to meet most analysts' needs. Because MixSIAR is open source and collaborative, we anticipate that new developments in mixing model methods, from parameterizations to model performance diagnostics, will continue to be incorporated into the functionality of MixSIAR. As such, the software provides a single tool that can meet the diverse needs of the rapidly increasing pool of stable isotope analysts, and affords developers a platform upon which to continue improving and diversifying mixing model analyses.

### Funding

Funding was provided in part by the Cooperative Institute for Marine Ecosystems and Climate (CIMEC) and the Center for the Advancement of Population Assessment Methodology (CAPAM). Brian C. Stock received support from the National Science

Foundation Graduate Research Fellowship under Grant No. DGE-1144086. There was no additional external funding received for this study. The funders had no role in study design, data collection and analysis, decision to publish, or preparation of the manuscript.

### Grant Disclosures

The following grant information was disclosed by the authors:
Cooperative Institute for Marine Ecosystems and Climate (CIMEC).
Center for the Advancement of Population Assessment Methodology (CAPAM).
National Science Foundation Graduate Research Fellowship: DGE-1144086.

### Competing Interests

Donald L. Phillips is the creator of EcoIsoMix.com, Corvallis, OR, USA.

### Author Contributions

- Brian C. Stock conceived and designed the experiments, performed the experiments, analyzed the data, prepared figures and/or tables, authored or reviewed drafts of the paper, approved the final draft.
- Andrew L. Jackson conceived and designed the experiments, authored or reviewed drafts of the paper, approved the final draft.
- Eric J. Ward conceived and designed the experiments, prepared figures and/or tables, authored or reviewed drafts of the paper, approved the final draft.
- Andrew C. Parnell conceived and designed the experiments, authored or reviewed drafts of the paper, approved the final draft.
- Donald L. Phillips conceived and designed the experiments, authored or reviewed drafts of the paper, approved the final draft.
- Brice X. Semmens conceived and designed the experiments, analyzed the data, prepared figures and/or tables, authored or reviewed drafts of the paper, approved the final draft.

### Data Availability

GitHub: https://github.com/brianstock/MixSIAR.
CRAN: https://CRAN.R-project.org/package=MixSIAR.

### Supplemental Information

Supplemental information for this article can be found online at http://dx.doi.org/10.7717/peerj.5096#supplemental-information.

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
