# Peer review of "Analyzing mixing systems using a new generation of Bayesian tracer mixing models"

_PeerJ, doi:10.7717/peerj.5096_

## Round 0.1 · original submission · Minor Revisions

Two reviewers have assessed this manuscript, and both found the manuscript to be well written with appropriate methodology and discussion of results. Both reviewers offer feedback for improving the manuscript, and I request that the authors address each comment from each reviewer during revision. One additional “review level” observation of my own is that in many locations I believe the text could use additional citations to support the claims that are made. For example, additional citations should be added to support the sentence on lines 52-55, perhaps including reference to paleoecological applications of isotope-based mixing models. A similar issue occurs on lines 68-73, 124-125, 140-142, 157-161, 252-254, and perhaps elsewhere.

·

Basic reporting

The manuscript is well written and I could not identify any major flaws or shortcomings.

Experimental design

Sound experimental design, good use of case study to illustrate the capabilities of the new mixing model formulation and platform.

Validity of the findings

As discussed in the article, mixing model analyses are becoming very common in ecological studies, however, the multitude of formulations complicates their use and interpretation. This article attempts to resolve some of the major issues in the field and provides a way forward.

Additional comments

Line 53 and throughout: The use of stable isotope “signatures” refers to the stable isotope composition of a significant reservoir like the geological mantle, air, ocean, lakes or a major part of the system being studied, not to the isotopic composition of individual samples. Most commonly, stable isotope composition of samples are referred to as stable isotope values. Consider revising.

I recommend this reference for stable isotope terminology: https://figshare.com/articles/Common_Mistakes_in_Stable_Isotope_Terminology_and_Phraseology/1150337


Line 148: No sure if the # is needed prior to (1). Revise if needed.

Line 151: Sort of a personal preference, but I suggest avoiding the use of and/or, as the / symbol should be relegated to mathematical equations. In most cases simply using “and” or “or” will work.

Line 153: Unless dissolved organic carbon and particulate organic matter is referenced in subsequent text there is no need to introduce the acronyms (DOC) and (POM).

Lines 155 and 156: See comment for line 53 regarding the use of “isotopic signatures”.

Line 165: Change (.) to (,) and remove # prior to (2).

Line 173: Not sure what “IID” means, not defined prior.

Line 175: Insert “,” after thus.

Line 176: See comment for line 53 regarding the use of “isotopic signatures”.

Line 181: Change (.) to (,) and remove # prior to (3).

Line 193: Not sure what “IID” means, not defined prior.

Line 212: See previous comment regarding “and/or” usage.

Line 226: Insert “,” after thus.

Line 235 and 245, throughout: Remove “/” in “uninformative”/generalist

Line 264: Insert “,” after equation before (4) and remove #.

Line 684: Citation for Nifong et al. (2015) used in the case study is not present in the References

·

Basic reporting

Well written and informative.

Experimental design

No comment.

Validity of the findings

The methodology is well explained and results are robust - with good discussion of the pitfalls.

Additional comments

A few general comments and suggestions:

Although I am enthusiastic about making Bayesian models more widely available and applicable, there are pitfalls, some of which are outlined in the manuscript, that will be difficult for researchers who are not familiar with Bayesian modeling to diagnose, let alone remedy. In that case, there is little substitute for experience. E.g., when is a lack of convergence due to mis-specification of priors, prior-data conflict, data issues, model-misspecification etc? This is often hard to diagnose, and perhaps it's worthwhile stating that reaching out to modelers (perhaps by mentioning the MixSIAR facebook, or pointing to github issues) is the best way to get over these hurdles. That may make it more appealing for people to pick up the tool. Issues such as those described in Brett (2014) could then be discussed and brought back to the information content of the data and prior choice.

L. 60: This fundamental mixing equation does not hold for fatty acid compositions if Y is the observed tracer value, as the measurement process for fatty acids leads to compositional data. Although the underlying (latent) mixing process may well conform to the equation, the observed outcomes do not. Given that the observed data for fatty acids is quite different from stable isotopes, I would suggest pointing out that the assumption of normality is not suitable for fatty acids as tracers, and that other software would need to be used (QFASA, fastinR). The ability to have covariates is also available in our fastinR package for fatty acids and stable isotopes (Neubauer & Jensen 2015), though admitted it has not been thoroughly tested (and hasn't been updated for a while).

L 264: The Dirichlet prior scaling: I think this is very difficult in practice – Fig 3 shows that although the unscaled prior from the actual observed prior sample size places very little weight on very small (near-zero) proportions for fish, the re-scaled prior places a lot of weight in that region, and comparatively little weight in the region of highest probability density of the original prior. This may lead to undesirable outcomes in practice. The text suggests the re-scaling provides a powerful tool and advocates running sensitivities to explore the impact of priors – although I generally agree, running sensitivities only seems useful if one understands what the prior assumptions are. Does MixSIAR provide ways to visualise the prior so that practitioners can understand what they are assuming for sensitivity runs? A sensible way to do this (and to deal with difficulties for linear model priors for mixing proportions) may be to let practitioners run their model without data to understand what the joint prior entails for mixing proportions. This may be especially relevant for models with covariates and/or random effects, the effects of which are not intuitive after transformation.

L338: Informative priors were discussed in the previous section.

---

## Round 0.2 · accepted · Accept

I appreciate the thorough job the authors did in considering the reviewer suggestions and in revising their manuscript.

#